# Biologically Inspired Dynamic Thresholds for Spiking Neural Networks

**Jianchuan Ding**[1]    **Bo Dong**[2]*    **Felix Heide**[2]
**Yufei Ding**[3]    **Yunduo Zhou**[1]    **Baocai Yin**[1]    **Xin Yang**[1]*
[1]Dalian University of Technology, Department of Computer Science
[2]Princeton University, Department of Computer Science
[3]University of California-Santa Barbara, Department of Computer Science

## Abstract

The dynamic membrane potential threshold, as one of the essential properties of a biological neuron, is a spontaneous regulation mechanism that maintains neuronal homeostasis, *i.e.*, the constant overall spiking firing rate of a neuron. As such, the neuron firing rate is regulated by a dynamic spiking threshold, which has been extensively studied in biology. Existing work in the machine learning community does not employ bioinspired spiking threshold schemes. This work aims at bridging this gap by introducing a novel bioinspired dynamic energy-temporal threshold (BDETT) scheme for spiking neural networks (SNNs). The proposed BDETT scheme mirrors two bioplausible observations: a dynamic threshold has 1) a positive correlation with the average membrane potential and 2) a negative correlation with the preceding rate of depolarization. We validate the effectiveness of the proposed BDETT on robot obstacle avoidance and continuous control tasks under both normal conditions and various degraded conditions, including noisy observations, weights, and dynamic environments. We find that the BDETT outperforms existing static and heuristic threshold approaches by significant margins in all tested conditions, and we confirm that the proposed bioinspired dynamic threshold scheme offers homeostasis to SNNs in complex real-world tasks.

## 1   Introduction

A spiking neural network (SNN) is a bioinspired neural network. Each spiking neuron is a mathematical model abstracted from the properties of a biological neuron. Spiking neurons communicate with each other through spike trains, mimicking the information transfer process of biological neurons [1, 2, 3]. Similar to how biological action potentials are all-or-none impulses, the spikes of SNNs are commonly binary voltage pulses. Leveraging this binary representation, specifically designed neuromorphic hardware [4, 5, 6], *e.g.*, TrueNorth [7] and Loihi [8], can run SNNs at extremely low power levels; they are 75 times more energy-efficient than their deep neural network counterparts on low-power GPU platforms [9]. As such, recently, SNNs have rapidly emerged as effective models for robotic control tasks, especially in mobile robots that demand low power consumption [10, 11].

However, existing SNNs suffer from poor generalizability, unlike their biological counterparts. Biologically, a neuron leverages a spontaneous regulation mechanism to maintain neuronal homeostasis [12]—the stable overall spiking firing rate or excitability within a network [13]—to robustly adapt to different external conditions and offer strong generalization. A dynamic threshold, one type of regulatory mechanism, plays an essential role in maintaining neuronal homeostasis by regulating the action potential firing rate; such thresholds are widely observed in different nervous systems [14, 15, 16, 17, 18, 19, 20, 21, 22, 23]. This threshold can be regarded as an adaptation to membrane potentials at short timescales [16], and it influences how the received signals of a neuron are encoded into a spike.

---

*Corresponding author `xinyang@dlut.edu.cn`; `bo.dong@princeton.edu`

36th Conference on Neural Information Processing Systems (NeurIPS 2022).

Even though different dynamic threshold schemes have been observed and extensively studied in neuroscience, only a handful of existing works investigate bioinspired dynamic threshold rules to improve the generalization of SNNs. Hao *et al.* [24] proposed a dynamic threshold method that relies on a heuristic dynamic scaling factor to gradually slow the growth of a threshold. Conversely, instead of controlling threshold growth, Shaban *et al.* [25] leveraged double exponential functions to manage the threshold decay. Kim *et al.* [26] used a predefined target firing count to adjust their threshold but did not define the optimal target firing count. No existing work has demonstrated that a bioinspired dynamic threshold scheme can achieve homeostasis in real-world tasks. More importantly, the existing work only validates the proposed dynamic threshold rules under ideal normal conditions without testing generalization to degraded conditions, which we argue is essential to validate whether homeostasis is achieved or not.

The direct use of bioplausible models in SNNs remains challenging, as most of these models are based on single cells in the nervous system and contain many optimized constants. In this work, we lift this limitation and introduce a novel dynamic energy-temporal threshold (BDETT) scheme for SNNs; the scheme comprises two components: a dynamic energy threshold and a dynamic temporal threshold schema. The two components reflect the following two biological observations: in *vivo*, the dynamic threshold exhibits a positive correlation with the average membrane potential and a negative correlation with the preceding rate of depolarization (*i.e.*, the excitatory status) [16]. The dynamic energy threshold is inspired by a biological predictive model which can predict the occurrences of spikes based on the previous membrane potential in the inferior colliculus of a barn owl [16]. The proposed dynamic temporal threshold component is inspired by the fact that a monoexponential function can effectively present a negative correlation [17, 22]. Notably, we provide an analysis of the original biological models and propose layerwise statistical cues for SNNs to replace the constants in the two original biological models.

We integrate the proposed BDETT into two widely used SNN models: a spike response model (SRM) [27] and a leaky integrate-and-fire (LIF) model [28]. The effectiveness of BDETT is validated with these two SNN models for autonomous robotic obstacle avoidance, continuous control and image classification tasks under normal and various degraded conditions, *e.g.*, dynamic obstacles, noisy inputs, and weight uncertainty. Extensive experimental results validate that the SNNs equipped with the proposed BDETT offer the strongest generalization across all tested scenarios. More importantly, we quantitatively validate that BDETT can significantly increase the homeostasis of the host SNN for robotic control tasks. This is the first work to demonstrate that dynamic threshold schemes can offer bioplausible homeostasis to SNNs in robotic real-world tasks under normal and degraded conditions, dramatically enhancing the generalizability and adaptability of the host SNNs.

In particular, we make the following contributions in this work:

- We introduce a bioinspired dynamic threshold scheme for SNNs that increases their generalizability.

- We devise a method that uses layerwise statistical cues of SNNs to set the parameters of our bioinspired threshold method.

- We validate that the proposed threshold scheme achieves bioplausible homeostasis, dramatically enhancing the generalizability across tasks, including obstacle avoidance and robotic control, and in normal and degraded conditions.

**Scope** We propose a novel approach to setting the parameters of our threshold scheme using layerwise statistical cues of an SNN. Although this is essential for the proposed method to be effective, implementing these statitical blocks directly in neuromorphic hardware may require extra engineering efforts, which is out of the scope of this work.

## 2 Background and Related Work

### 2.1 Spiking Neural Networks (SNNs)

Various models for spiking neurons have been described to mathematically describe the properties of a nervous neuron. Typically, three conditions are considered by these models: resting, depolarization, and hyperpolarization. When a neuron is resting, it maintains a constant membrane potential. The change in membrane potential can be either a decrease or an increase relative to the resting potential. An increase in the membrane potential is called depolarization, which enhances the ability of a cell to generate an action potential; it is excitatory. In contrast, hyperpolarization describes a reduction in

the membrane potential, which makes the associated cell less likely to generate an action potential, and, as such, is inhibitory. All inputs and outputs of a spiking neuron model are sequences of spikes. A sequence of spikes is called a spike train and is defined as $s(t) = \Sigma_{t^{(f)} \in \mathcal{F}} \delta(t - t^{(f)})$, where $\mathcal{F}$ represents the set of times at which the individual spikes occur [29]. Typical spiking neuron models set the resting potential as 0. However, existing models achieve depolarization and hyperpolarization in substantially different ways. In the following, we briefly review two commonly used models: the spike response model (SRM) [27] and leaky integrate-and-fire (LIF) model [28]. More details about these two models are provided in Supplementary Note 1.

**Spike Response Model (SRM)** An SRM first converts an incoming spike train $s_i(t)$ into a spike response signal as $(\varepsilon * s_i)(t)$, where $\varepsilon(\cdot)$ is a spike response kernel. Then, the generated spike response signal is scaled by a synaptic weight $w_i$. Depolarization is achieved by summing all the scaled spike response signals: $\Sigma_i w_i (\varepsilon * s_i)(t)$. When incoming spike trains trigger a spike $s(t)$, the SRM models hyperpolarization by defining a refractory potential as $(\zeta * s)(t)$, where $\zeta(\cdot)$ is a refractory kernel.

**Leaky Integrate-and-Fire (LIF)** An LIF model is a simplified variant of an SRM. This scheme directly processes incoming spike trains and ignores the spike response kernel. Hyperpolarization is achieved by a simplified step decay function, $f_d(s(t)) = D$ for $s(t) = 0$; 0 for $s(t) = 1$.

## 2.2 Spiking Neural Networks for Robot Control

Biological neural circuits have an impressive ability to avoid obstacles robustly in complex dynamical environments, *e.g.*, as in dragonfly flight trajectories. Inspired by this observation, recently, researchers have explored SNNs for obstacle avoidance [30, 31, 32, 33]. For example, Tang *et al.* [33] devised an SNN to mimic a neurophysiologically plausible connectome of the brain's navigational system without assuming all-to-all connectivity. Following the path, Tang *et al.* [9] proposed a spiking deep deterministic policy gradient (SDDPG) method to train a LIF-based spiking actor-network (SAN) for mapless navigation. They show that SNNs can robustly control a robot in mapping tasks while being able to explore an unknown environment. SNNs have also been proposed for continuous robot control tasks. Patel *et al.* [34] proposed to combine SNNs with a Deep Q-network algorithm, improving the robustness to occlusion in the input image. Tang *et al.* [35] proposed a population-coded spiking actor network (PopSAN) to solve high-dimensional continuous control problems, trained using deep reinforcement learning algorithms. Recently, modern neuromorphic hardware has made it possible to deploy SNNs on neuromorphic processors in ultra-low power envelopes [9, 36, 37, 38]. Compared to existing convolutional deep policy networks [39] on the mobile-GPUs such as the Nvidia Jetson TX2, SAN and PopSAN on Loihi neuromorphic processor consume 75 and 140 times less energy per inferences, respectively. All SNN-based models discussed above only consider static spiking thresholds. More importantly, experiments show that they suffer from poor generalization and fail in realistic degraded conditions. In this work, we use both SAN and PopSAN as testbeds and baseline methods to validate the effectiveness of the proposed bioinspired dynamic threshold scheme, BDETT.

## 3 Bioinspired Dynamic Energy-Temporal Threshold (BDETT)

Motivated by the behavior of spiking threshold dynamics in biological nervous systems, we propose a model with dynamic thresholds that exhibit positive and negative correlations with the average membrane potential and the preceding rate of depolarization, respectively. To achieve this behavior in the proposed scheme, given the $i$-th neuron in the $l$-th layer at timestamp $t + 1$, we define a dynamic threshold $\Theta_i^l(t + 1)$ as

$$\Theta_i^l(t + 1) = \frac{1}{2}(\mathrm{E}_i^l(t) + \mathrm{T}_i^l(t + 1)), \tag{1}$$

where $E_i^l(t)$ is the dynamic energy threshold (DET) of the neuron for ensuring a positive correlation, and $T_i^l(t + 1)$ is the dynamic temporal threshold (DTT), which ensures a negative correlation; see Figure 1a. Note that each neuron has a different dynamic threshold at timestamp $t + 1$ based on the proposed DET and DTT, which we describe below.

**Dynamic Energy Threshold (DET)** Positive correlations between dynamic thresholds and average membrane potentials have been observed in several areas of diverse biological nervous systems, such as the visual cortex and auditory midbrain [17, 21, 22]. With sufficient voltage measurements at spike onsets, one can fit a model to directly predict the voltage of a threshold [40]. However, the fitted biological model is only meaningful to a specific nervous system, and stimulus or measurement uncertainty can significantly impact the model accuracy. Fontaine *et al.* [16] proposed a biological predictive approach to assess the occurrence of spikes based on the previous membrane potential; this

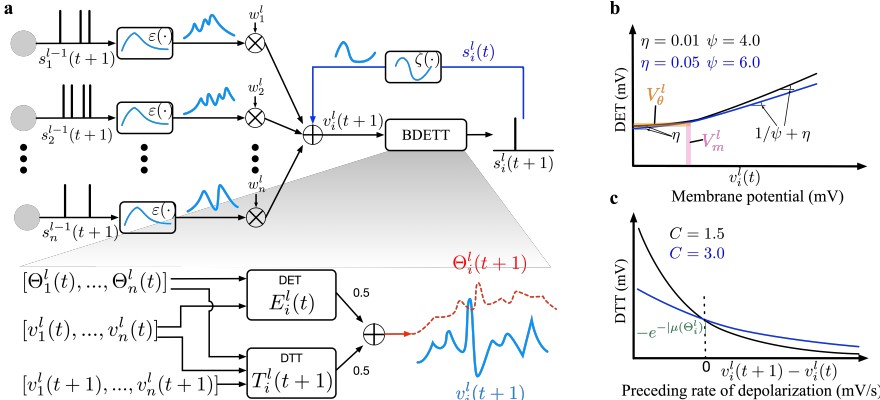

Figure 1: **An illustration of the proposed BDETT scheme**. a. We demonstrate the intuitive idea of our BDETT scheme for the $i$-th neuron in the $l$-th layer at timestamp $t + 1$ from the perspective of an SRM-based SNN model; $\Theta_i^l(t)$ and $v_i^l(t)$ are the dynamic threshold and postsynaptic membrane potential of the $i$-th neuron in the $l$-th layer at timestamp $t$, respectively. b & c. Two example DET and DTT graphs, respectively.

method does not rely on voltage measurements at spike onsets. Even though the model was based on a barn owl's inferior colliculus, it exhibits great generality in terms of threshold variability statistics with other nervous systems (*e.g.*, cortical neurons) [16]. The proposed dynamic energy threshold is inspired by this biological predictive model but includes several changes that are critical for the model to be effective in SNNs. For the $i$-th neuron in the $l$-th layer at timestamp $t$, we define

$$\mathrm{E}_i^l(t) = \eta(v_i^l(t) - V_m^l(t)) + V_\theta^l(t) + ln(1 + e^{\frac{v_i^l(t) - V_m^l(t)}{\psi}}), \tag{2}$$

$$V_m^l(t) = \mu(v_i^l(t)) - 0.2(\max(v_i^l(t)) - \min(v_i^l(t))) \quad \text{for } i = 1, 2, ..., n^l, \tag{3}$$

$$V_\theta^l(t) = \mu(\Theta_i^l(t)) - 0.2(\max(\Theta_i^l(t)) - \min(\Theta_i^l(t))) \quad \text{for } i = 1, 2, ..., n^l, \tag{4}$$

where $v_i^l(t)$ is the neuron postsynaptic membrane potential at timestamp $t$; $\mu$ is the mean operator; $n^l$ is the total number of neurons in the $l$-th layer; and $\eta$ and $\psi$ are two hyperparameters, which are set empirically. Figure 1b shows two example graphs for Eq. 2; $\eta$ controls the shallow slope, and $\frac{1}{\psi} + \eta$ defines the slope of the steep part.

Intuitively, $V_\theta^l(t)$ and $V_m^l(t)$ define a critical region. When the membrane potential $v_i^l(t)$ is smaller than $V_m^l(t)$, the function has a shallower slope, and the threshold value is dominated by $V_\theta^l(t)$. In the opposite case, the energy threshold has a higher rate of increase to inhibit a high spiking firing rate. In the biological predictive model proposed by Fontaine et al. [16], $V_m^l(t)$ and $V_\theta^l(t)$ are the constants to be optimized during the model fitting process. However, we find that directly adopting these two fitted constants in an SNN does not result in generalization; see section 4.4. To tackle this challenge, we leverage the statistical cues of SNN layers to adjust these two important parameters, as defined in Eqs. 3 and 4. Specifically, we model $V_m^l(t)$ as the mean of the membrane potentials of the neurons in the layer $l$. The mean value is shifted by a bias, $0.2(\max(v_i^l(t)) - \min(v_i^l(t)))$, which is based on the range of the potentials; see Eq. 3. The motivation behind this formulation is that we aim to couple the DET and the potentials of all other neurons in the same layer. Furthermore, we leverage the bias term to adjust the DET sensitivity to the layerwise potential range. Here, $V_\theta^l(t)$ is modeled based on similar insights, where we use threshold potentials (*i.e.*, $\Theta_i^l(t)$) instead of membrane potentials; see Eq. 4. We note that the performance of the proposed BDETT is not sensitive to the constant value 0.2; see Supplementary Note 7 for details.

**Dynamic Temporal Threshold (DTT)** We propose a DTT scheme to address the observed negative correlation between the spiking threshold and the preceding rate of depolarization. Azouz *et al.* [17, 22] discovered that a monoexponential function $y = a + be^{-V/C}$ can effectively capture the negative correlation of a biological neuron, where $V = dV_m/dt$; $C$ is a decay constant; and $a$, $b$, and $C$ are parameters to optimize. The authors applied this function to 42 cortical neurons and found significant correlations in 92% of the trials [17]. We propose a variant of this mechanism. In particular, we replace the constant $a$ with an exponential decay function, and we base the decay rate on the mean of the dynamic thresholds of all neurons in the $l$-th layer at the previous timestamp $t$; $b$ is set to 1. Additionally, we empirically set the delay constant $C$. Mathematically, for the $i$-th neuron

in the $l$-th layer, the DTT at timestamp $t + 1$ is defined as

$$\text{T}_i^l(t + 1) = a + e^{\frac{-(v_i^l(t+1) - v_i^l(t))}{C}}, \tag{5}$$

$$a = -e^{-|\mu(\Theta_i^l(t))|} \quad \text{for } i = 1, 2, ..., n^l. \tag{6}$$

Figure 1c shows two example graphs for Eq. 5. These plots highlight that higher depolarization (*i.e.*, $v_i^l(t + 1) - v_i^l(t) > 0$) leads to a lower temporal threshold, while higher hyperpolarization (*i.e.*, $v_i^l(t + 1) - v_i^l(t) < 0$) significantly increases the temporal threshold. We propose modeling $a$ similar to how $V_m^l(t)$ is modeled in the DET, that is, by coupling the DTT value and the layerwise dynamic thresholds at the previous timestamp $t$ (*i.e.*, $\Theta_i^l(t)$). The delay constant $C$ adjusts the sensitivity of the DTT to changes in the temporal potential of a neuron. As shown in Figure 1c, a lower $C$ value results in a substantially faster drop in the DTT value (*i.e.*, the black curve) than that provided by a higher $C$ value (*i.e.*, the blue curve).

**Interaction of DET and DTT** A critical difference between DET and DTT lies in the drivers of the two threshold schemes. DET leverages the magnitude of the membrane potential to estimate a threshold, while the DTT based on the preceding rate of depolarization. Therefore, they may be counteracting or helping each other to achieve an optimal threshold. One example is that when noise causes low potential fluctuations, the overall threshold should increase to suppress the noise. In this case, the DET increases as the noise increases the membrane potential. However, DTT remains at a relatively constant threshold (*i.e.*, $a + 1$) as the preceding rate of depolarization caused by the noise is close to 0. When a neuron experiences a fast membrane potential drop, *e.g.*, during the relative refractory period, we expect the overall threshold to increase. In this scenario, even though DET decreases with the reduced membrane potential, DTT increases faster. Hence, the proposed method increases the overall threshold in this case. Please see Supplementary Note 10 for details.

## 4 Experiments

We assess the effectiveness of BDETT on three different tasks: robot obstacle avoidance, robotic continuous control and image classification. In the robot obstacle avoidance task, a robot aims to reach a randomly chosen destination without touching any obstacle within 1000 steps, counted as a "pass". For this task, we assess methods by measuring success rate (SR), the percentage of successful passes out of 200 trials. As continuous control tasks, we evaluate the HalfCheetah-v3 and Ant-v3 control outputs (see Figure 3a) from the OpenAI gym [41]. In these two continuous control tasks, an agent relies on a learned SNN-based control policy to decide the next action based on the current observation (*i.e.*, state), and each action is associated with a reward; see Figure 3a. We assess control policies with the total sum of the rewards. Note that the Ant-v3 control task is more challenging than HalfCheetah-v3, with significantly large state and action spaces. Top-1 classification accuracy is used to assess image classification.

For the robotic control tasks, in addition to evaluating the control output, *i.e.*, SR and total reward, we also measure the homeostasis of the host SNNs. In particular, we use three statistical metrics, $\text{FR}_m$, $\text{FR}_{std}^m$, and $\text{FR}_{std}^s$, to quantify the homeostasis of an SNN; these metrics are based on the neuron firing rate. $\text{FR}_m$ is the mean neuron firing rate of an SNN across all $P$ trials; $\text{FR}_{std}^m$ is the average of $P$ standard deviations, and each of them is the standard deviation of the neuron firing rates of an SNN during a single trial; $\text{FR}_{std}^s$ denotes the standard deviation of the $P$ standard deviations. $\text{FR}_{std}^s$ represents the standard deviation across all $P$ trials, while $\text{FR}_{std}^m$ denotes the mean of these standard deviations. Details on these three metrics can be found in Supplementary Note 2.

**Experimental Setup** For robot obstacle avoidance tasks, the we use variants of the spiking actor network (SAN) [9] as host SNN. The original SAN uses LIF as its neuron model, but it resets the membrane potentials of all neurons to zero for each robot state. The resting operation is contradictory to the leaky function of LIF. Therefore, we modify the SAN by removing the resting operation, which is dubbed SAN-NR. To validate the effectiveness of the proposed BDETT, we integrate it into both LIF-based and SRM-based SAN-NR models and compare them with their original static threshold and two heuristic dynamic threshold schemes, DT1 [24] and DT2 [26]. See Supplementary Note 2 for details on the DT1 and DT2 schemes. We set the batch size to 256 and the learning rate to 0.00001 for both the actor and critic networks during the training process. In addition, we use the following hyperparameter settings for the proposed BDETT: $\eta = 0.01$ and $\psi = 4.0$ for the DET and $C = 3.0$ for the DTT. For estimating homeostasis, we set $P = 200$. See Supplementary Notes 4 for further training details.

For robot continuous control, we adopt the population-coded SAN (PopSAN) [35] as our baseline model; it is a modified version of SAN [9] with a specifically designed encoder and decoder for accommodating high-dimensional control tasks. Note that PopSAN does not rest the membrane potentials as the encoder leverages soft-reset IF neurons. Hence, PopSAN is the counterpart of the SAN-NR used in the obstacle avoidance tasks. We integrate BDETT into both LIF- and SRM-based PopSAN models and compare them with their original static threshold schemes and the two heuristic dynamic schemes, DT1 and DT2. Following the evaluation settings of PopSAN [35], we train ten models corresponding to ten random seeds, and the best-performing model is used for our assessment conducted under different degraded conditions. In particular, the best-performing model is evaluated ten times under each experimental condition, and the mean reward of the ten evaluations represents the model performance. Each evaluation consists of ten episodes, and each episode lasts for a maximum of 1000 execution steps. Hence, the $P$ value used for estimating homeostasis is set to 100, *i.e.*, 10 episodes $\times$ 10 evaluations. PopSAN and its variants are trained by using the twin-delayed deep deterministic policy gradient off-policy algorithm [42]. The hyperparameter settings of BDETT is the same as the ones used for obstacle avoidance tasks, except the $\psi$ for the DET is set to 6.0. Following the training protocol of PopSAN [35], we set the batch size to 100 and the learning rate to 0.0001 for both the actor and critic networks. The reward discount factor is set to 0.99, and the maximum length of the replay buffer is set to 1 million. See Supplementary Notes 5 for training details.

For the SRM-based baseline methods, the spike response kernel and refractory kernel of the SRM are adopted from [27, 29], and they are defined as $\varepsilon(t) = te^{1-t}$ and $\zeta(t) = -2\Theta(t)e^{-t}$, respectively. For all tasks, each dimension of a robot state is encoded into a spike train with $T$ timesteps. All experimental results are obtained with $T = 5$. For a demonstration of the generalization provided by the BDETT, we provide the experimental results obtained with $T = 25$ in Supplementary Notes 4, 5, 6 for the obstacle avoidance, HalfCheetah-v3, and Ant-v3 tasks, respectively.

## 4.1 Robot Obstacle Avoidance with BDETT

We evaluate the proposed method for robot obstacle avoidance tasks with one standard condition, *i.e.*, static obstacles, and three specifically designed adverse conditions: dynamic obstacles, degraded inputs, and weight uncertainty. For the dynamic obstacle experiments, we introduce 11 dynamically moving cylinders in a static testing environment, and each repeatedly wanders between two points; see Figure 2a. The wandering distance and speed are designed to provide sufficient space and time to allow possible passes. The robot utilizes a Robo Peak light detection and ranging (RPLIDAR) system as its sensing device to detect obstacles, offering a field of view of 180 degrees with 18 range measurements, as shown in Figure 2b.

In our degraded input scenario, we disturb the obtained range measurements in three different ways: "0.2": We set the range of the 3rd, 9th, and 15th lasers to 0.2 m, always reporting obstacles even when none occur; "6.0": This is similar to the "0.2" setting, but we set the three lasers' ranges to 6.0 m, which is the average visible range in the test environment and means that the three lasers cannot perceive any obstacles; "GN": We add Gaussian noise [43] to each of the 18 range measurements. The three proposed degraded input settings are illustrated in Figure 2c.

In the weight uncertainty experiments, as illustrated in Figure 2d, the learned synaptic weights of the host SNNs are also disturbed in three different ways. "8-bit Loihi weight": Neuromorphic hardware (*e.g.*, Loihi) achieves computing efficiency by sacrificing the weight precision. Therefore, when deploying an SNN on neuromorphic hardware, one needs to scale and round up the learned floating-point synaptic weights to low-precision weights. "GN weight": We add Gaussian noise, $\mathcal{N}(0, 0.05)$, to all synaptic weights. "30% zero weight": Among the synaptic weights between every two adjacent layers, we randomly set 30% of them to 0. To reduce the impact of the randomness introduced in the "GN weight" and "30% zero weight" experiments, we report the average success rates (SRs) and standard deviation of 5-round tests.

**Success Rate** The SRs of the competing LIF- and SRM-based approaches across all experimental settings are reported in Figure 2e and Table 1. For the "GN weight" and "30% zero weight" experiments, the standard deviations of the 5-round SRs are also reported. The proposed BDETT achieves the highest SRs in all experiments, demonstrating its effectiveness. Notably, under dynamic obstacle conditions, the BDETT outperforms the runners-up by significant margins (9% versus the LIF and 12% versus the SRM). Under degraded input conditions, the BDETT yields at least 10% more successful passes than other competing methods. In the weight uncertainty experiments, our BDETT increases the SRs of the baseline SAN-NR model by at least 10.5%, 24.6%, and 15.6% under "8-bit Loihi weight", "GN weight", and "30% zero weight" settings, respectively. We observe

that our BDETT can help the robots effectively avoid both static and dynamic obstacles under all three adverse conditions; see Supplementary Tables 2, 3, and 4 for details.

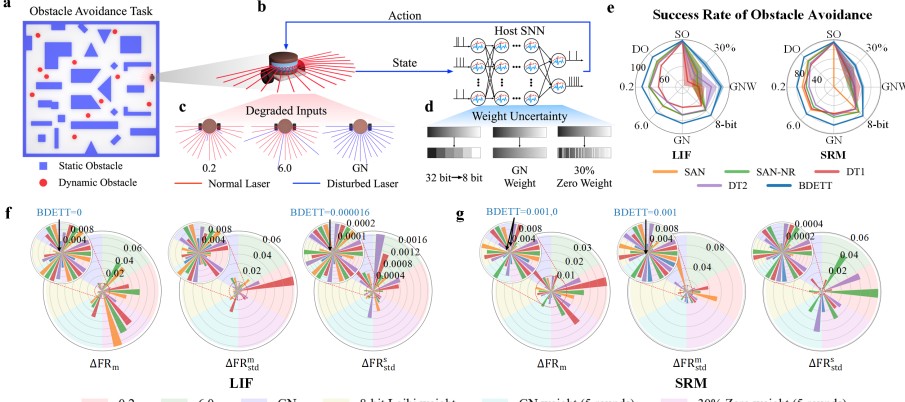

Figure 2: Proposed method for robot obstacle avoidance. a. The static and dynamic testing environments of the obstacle avoidance tasks. b. The control loop of a robot. c. The three specifically designed degraded input conditions. d. A demonstration of the three weight uncertainty experimental settings. e. The SRs of obstacle avoidance under all experimental settings. 'SO' and 'DO' indicate the testing environments with static and dynamic obstacles, respectively; '30%' and 'GNW' denote the "30% zero weight" and "GN weight" conditions. f & g show the LIF- and SRM-based SNNs' homeostasis changes with respect to the base condition (*i.e.*, DO) in terms of three metrics. e-h use the same color codes as shown in f.

Table 1: **Quantitative performance of obstacle avoidance under degraded conditions.** Here, $\sigma$ is the standard deviations of the 5-round SRs.

| Type | Name | LIF SR↑ | SRM SR↑ | Type | Name | LIF SR↑ | SRM SR↑ | Type | Name | LIF SR↑ | SRM SR↑ |
|---|---|---|---|---|---|---|---|---|---|---|---|
| 0.2 | SAN | 78.5% | 68% | 6.0 | SAN | 71% | 70% | GN | SAN | 71.5% | 57% |
| | SAN-NR | 80% | 59% | | SAN-NR | 70% | 61.5% | | SAN-NR | 72% | 65.5% |
| | DT1 [24] | 65.5% | 64% | | DT1 [24] | 62% | 67% | | DT1 [24] | 60.5% | 58% |
| | DT2 [26] | 78% | 53.5% | | DT2 [26] | 61.5% | 55% | | DT2 [26] | 71.5% | 61.5% |
| | BDETT | **90%** | **79.5%** | | BDETT | **84.5%** | **83%** | | BDETT | **84.5%** | **82.5%** |
| 8-bit Loihi weight | SAN | 78.5% | 77% | GN weight (5 rounds) | SAN | 51.3% ($\sigma$-6.8) | 0% ($\sigma$-0) | 30% Zero weight (5 rounds) | SAN | 59.3% ($\sigma$-10.5) | 0% ($\sigma$-0) |
| | SAN-NR | 79.5% | 76.5% | | SAN-NR | 52.5% ($\sigma$-7.1) | 37.2% ($\sigma$-7.6) | | SAN-NR | 61.6% ($\sigma$-7.5) | 46.5% ($\sigma$-12.4) |
| | DT1 [24] | 70% | 67% | | DT1 [24] | 54.6% ($\sigma$-7.9) | 44.9% ($\sigma$-11.4) | | DT1 [24] | 41.2% ($\sigma$-7.7) | 44.3% ($\sigma$-11.7) |
| | DT2 [26] | 78.5% | 67.5% | | DT2 [26] | 73.2% ($\sigma$-7.4) | 43.6% ($\sigma$-4.4) | | DT2 [26] | 55.6% ($\sigma$-9.3) | 49.1% ($\sigma$-10.8) |
| | BDETT | **90%** | **88.5%** | | BDETT | **87.7% ($\sigma$-3.3)** | **61.8% ($\sigma$-2.9)** | | BDETT | **77.2% ($\sigma$-3.6)** | **65.2% ($\sigma$-2.7)** |

**Homeostatic Evaluation** When an SNN is in homeostasis, all neurons are expected to have similar and sparse firing patterns under different conditions [44, 45]. Therefore, when transferring from one condition to another, the SNNs with stronger homeostasis are expected to induce fewer changes in all three metrics. The changes induced in all successful trials involving the LIF- and SRM-based host SNNs under different experimental settings are illustrated in Figures 2f and g, respectively. The changes (*i.e.*, in $\Delta\text{FR}_m$, $\Delta\text{FR}_{std}^m$, and $\Delta\text{FR}_{std}^s$) are estimated with respect to the corresponding homeostasis achieved in the dynamic obstacle experiments, *i.e.*, under the base condition. The proposed BDETT scheme yields minimal changes in all three metrics when transferring from the base condition to all other experimental settings, except for the $\Delta\text{FR}_{std}^s$ estimated based on the SRM-based 8-bit Loihi weight experiment. The figures highlight that the proposed BDETT significantly improves on the baseline SAN-NR model, as evidenced by the remarkable drops in these three statistical metrics. For example, as shown in the "6.0" section of the $\Delta\text{FR}_m$ in Figure 2f, our dynamic threshold scheme reduces the $\Delta\text{FR}_m$ from 0.043 to 0.001. In the "0.2" section of the $\Delta\text{FR}_{std}^s$ in Figure 2g, the $\Delta\text{FR}_{std}^s$ is decreased to 1.7% of its original value (from 0.0058 to 0.0001). We also witness that the DT1 and DT2 schemes significantly weaken the baseline model's homeostasis, as shown in Figure 2f in the "0.2" section of the $\Delta\text{FR}_{std}^m$ and the "6.0" section of the $\Delta\text{FR}_{std}^s$.

The goal of homeostasis to enhance the host SNN generalization. Therefore, we expect SNNs with stronger homeostasis (*i.e.*, smaller $\Delta\text{FR}_m$, $\Delta\text{FR}_{std}^m$, and $\Delta\text{FR}_{std}^s$ values) to outperform those with weaker homeostasis. Our experimental results confirm this, validating that the strong homeostasis provided by our BDETT can improve the generalization capabilities of SNNs to different degraded

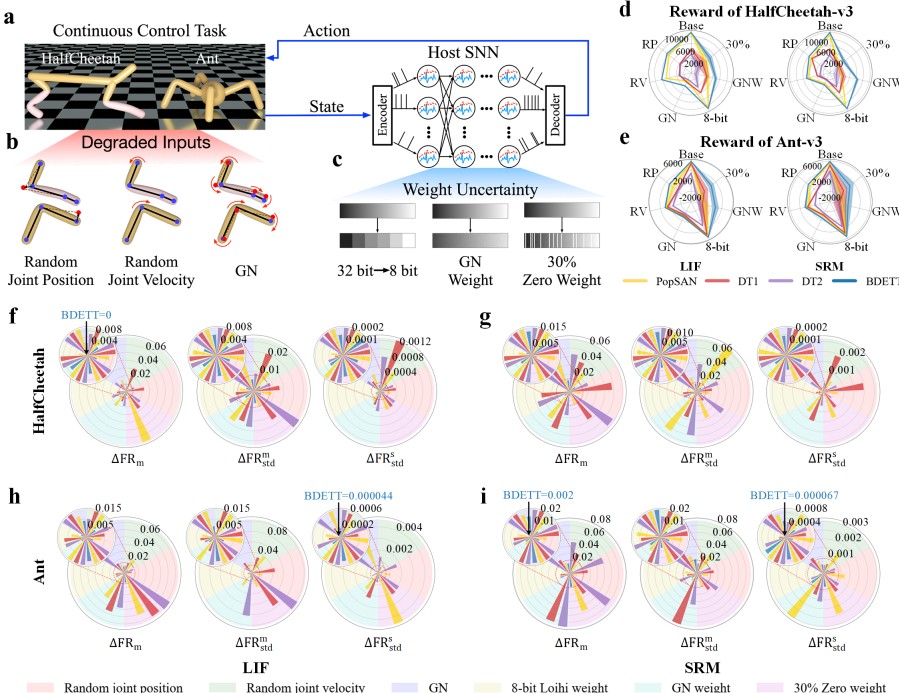

Figure 3: Proposed method for continuous robot control. a. The control loops of HalfCheetah-v3 and Ant-v3. b. Examples of the three specifically designed degraded input conditions, where the red dots and curved arrows indicate the disturbed joint positions and velocities, respectively. c. The three specifically designed weight uncertainty conditions. d & e. The rewards of the HalfCheetah-v3 and Ant-v3 tasks across all experimental conditions, respectively. 'Base' indicates the normal base condition; 'RP' and 'RV' denote 'Random joint position' and 'Random joint velocity'. f & g. The LIF- and SRM-based SNNs' homeostasis changes with respect to the 'Base' condition in the HalfCheetah-v3 tasks. h & i. The LIF- and SRM-based SNNs' homeostasis changes with respect to the 'Base' condition in the Ant-v3 tasks. **d-i** use the same color codes shown in e.

conditions. We argue that this is a highly desired capability not only for mobile robotics but also for broader machine learning. See Supplementary Note 4 for more experimental results and analysis.

## 4.2 Continuous Robot Control with BDETT

For the HalfCheetah-v3 and Ant-v3 tasks, similar to the robot obstacle avoidance tasks, we evaluate on one standard and two specifically designed degraded inputs and weight uncertainty adverse conditions to demonstrate the strong generalization enabled by our BDETT. In this context, for the degraded input conditions, we disturb the observations of these two control tasks in three ways. "Random joint position": For each episode, one of the joint positions is randomly selected, and its original position is replaced by a random number sampled from a Gaussian distribution $\mathcal{N}(0, 0.1)$. "Random joint velocity": We randomly select one of the joint velocities in each episode and change its observed velocity to a random number sampled from a Gaussian distribution $\mathcal{N}(0, 10.0)$. "GN": In each episode, we add Gaussian noise sampled from the distribution $\mathcal{N}(0, 1.0)$ to each dimension of a state; see Figure 3b. The weight uncertainty conditions of the control tasks are the same as those used in the robot obstacle avoidance tasks, as illustrated in Figure 3c.

**Rewards** As shown in Figures 3d, e and Table 2, under all experimental settings, the proposed BDETT offers the host SNNs the highest rewards, significantly improving upon the rewards of the baseline PopSAN model by at least $438$ (*i.e.*, the SRM-based PopSAN model under the "GN" setting) for the HalfCheetah-v3 tasks and $213$ (*i.e.*, the LIF-based PopSAN model under the "Random joint velocity" setting) for the Ant-v3 tasks. Notably, under weight uncertainty conditions with a HalfCheetah-v3 agent, even with low-precision 8-bit weights, the proposed BDETT helps the SRM-based host SNN achieve a higher reward than that obtained with high-precision floating-point weights (11767 vs. 11268); see Supplementary Tables 7 and 9. With an Ant-v3 agent, the proposed BDETT helps both the LIF- and SRM-based host SNNs achieve higher rewards, even with low-precision weights, *i.e.*, 5570 vs. 5526 and 5648 vs. 5643, respectively. See Supplementary Notes 5 and 6

Table 2: **Quantitative performance of continuous robot control tasks under degraded conditions.** For each cell, we report the estimated rewards for both HalfCheetah-v3 and Ant-v3 tasks in the format of HalfCheetah-v3/Ant-v3.

| | | LIF | SRM | | | LIF | SRM | | | LIF | SRM |
|---|---|---|---|---|---|---|---|---|---|---|---|
| **Type** | **Name** | Reward↑ | Reward↑ | **Type** | **Name** | Reward↑ | Reward↑ | **Type** | **Name** | Reward↑ | Reward↑ |
| Random joint position | PopSAN | 7832/2503 | 7120/3004 | Random joint velocity | PopSAN | 7020/2890 | 6576/2372 | GN | PopSAN | 2440/977 | 3457/1031 |
| | DT1 [24] | 3923/1435 | 6830/1333 | | DT1 [24] | 3187/2628 | 3836/2508 | | DT1 [24] | 2790/922 | 2210/958 |
| | DT2 [26] | 3750/1280 | 3230/1330 | | DT2 [26] | 3395/1579 | 3031/1025 | | DT2 [26] | 1994/560 | 2307/583 |
| | BDETT | **8465/3339** | **7883/3450** | | BDETT | **8302/3103** | **7116/2984** | | BDETT | **3909/1269** | **3895/1559** |
| 8-bit Loihi weight | PopSAN | 10728/5347 | 10802/5285 | GN weight | PopSAN | 4640/637 | 3583/467 | 30% Zero weight | PopSAN | 5020/287 | 3233/372 |
| | DT1 [24] | 6026/5004 | 6569/4889 | | DT1 [24] | 4483/221 | 4128/-57 | | DT1 [24] | 3995/1247 | 3503/1450 |
| | DT2 [26] | 4372/3122 | 4629/3463 | | DT2 [26] | 1334/-265 | 2028/-173 | | DT2 [26] | 2721/-548 | 3056/-203 |
| | BDETT | **10823/5570** | **11767/5648** | | BDETT | **6928/2782** | **8381/1658** | | BDETT | **6551/2931** | **5386/3046** |

for additional experimental results and analysis related to the HalfCheetah-v3 and Ant-v3 tasks, respectively.

**Homeostatic Evaluation** In Figures 3f-i, we show the changes induced in these three metrics when shifting from normal conditions (*i.e.*, the base conditions) to all other experimental settings. The proposed BDETT offers the strongest homeostasis to the host SNNs among all competing approaches for both the HalfCheetah-v3 and Ant-v3 control tasks. In particular, for the HalfCheetah-v3 control task, as shown in the "30% zero weight" section of the $\Delta FR_m$ in Figure 3f, our dynamic threshold scheme reduces the $\Delta FR_m$ of the baseline PopSAN model from 0.069 to 0.006. In the "GN weight" section of the $\Delta FR^s_{std}$ in Figure 3g, the proposed BDETT decreases the $\Delta FR^s_{std}$ of the SRM-based PopSAN to $8.3\%$ of its original value (from 0.0012 to 0.0001); see Supplementary Table 10 for details. For the Ant-v3 control task, as shown in the "GN weight" section of the $\Delta FR_m$ in Figure 3h, our dynamic threshold scheme reduces the $\Delta FR_m$ of the LIF-based baseline model from 0.041 to 0.003. In the "Random joint position" section of the $\Delta FR^s_{std}$ in Figure 3h, the $\Delta FR^s_{std}$ of the LIF-based baseline model is decreased to $10\%$ of its original value (from 0.0010 to 0.0001); see Supplementary Table 15 for details. As in the obstacle avoidance tasks, the DT1 and DT2 schemes significantly decrease the homeostasis of both the LIF- and SRM-based baseline models in both continuous control tasks. Some extreme cases are shown in the "Random joint velocity" section of the $\Delta FR^m_{std}$ in Figure 3f, and the "GN weight" section of the $\Delta FR_m$ in Figure 3i.

These experimental results obtained for the two continuous control tasks support the observations obtained in the obstacle avoidance tasks. More importantly, we witness that the strong homeostasis provided by our BDETT improves generalization to severely degraded conditions.

### 4.3 Image Classification with BDETT

We assess the proposed SNN-based learning method on image classification as a relevant vision task. In particular, to measure the generalization of the proposed BDETTscheme, we conduct additional image classification experiments under normal and degraded conditions. To this end, we simulate degraded inputs similar to the robotic control tasks; see the weight uncertainty degraded settings illustrated in Figure 4d. In addition, we test on degradations that are tailored to classification from two adversarial attack methods; the fast gradient sign method (FGSM) [46] and projected gradient descent (PGD) [47]; see Figure 4c.

Specifically, we train the SCNN model [48] on the MNIST dataset [49] as our baseline model. Each pixel of an MNIST image is encoded into 30 Poisson spikes as inputs to SCNN for training and testing. As shown in Figure 4e and Table 3, directly applying the proposed approach without any changes to image classification in degraded conditions compares favorably across all experimental settings and in terms of generalization. For stronger degradations, the Top-1 classification accuracy of both the baseline and our approach decreases, but the proposed method is less affected. Note that the SCNN model contains CNN layers, blocking us from estimating homeostasis.

### 4.4 BDETT without Statistical Parameter Adjustment

We found it essential to replace the constants in the two biological models we base our approach on with layerwise statistical cues. Here, we report the performance of the BDETT with the original constants of the fitted biological models, demonstrating the effectiveness of the proposed layerwise statistical parameter settings. In particular, we first use the corresponding constants of the fitted adaptive threshold model [16] and replaced the $V^l_m(t)$ and $V^l_\theta(t)$, *i.e.*, Eq. 3 and Eq. 4, with 3 and 7, respectively. These two constants are obtained by shifting the originally fitted constants $-67$ and $-63$ by 70 to compensate for the difference of the rest potentials; $-70$ mV in the original model but 0 mV

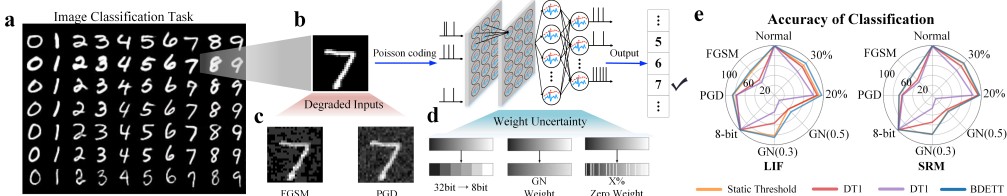

Figure 4: Proposed method for image classification. a. The examples of MNIST dataset. b. The forward pass of SCNN model [48]. c. Examples of the two adversarial samples as degraded input conditions. d. The three specifically designed weight uncertainty conditions. e. The Top-1 accuracy(AC) of image classification under all condition settings. '20%' and '30%' denote the "20% zero weight" and "30% zero weight" condition, respectively; 'GN(x)' denotes the Gaussian noise, $\mathcal{N}(0, x)$, in "GN Weight" settings.

Table 3: **Quantitative performance of image classification tasks in Top-1 accuracy (AC).** For FGSM, we set $\epsilon = 0.2$. For PGD, we set $\epsilon = 0.01$ and run 20 iterations. GN(x) indicate Gasussian noise, $\mathcal{N}(0, x)$.

| Type | Name | LIF AC↑ | SRM AC↑ | Type | Name | LIF AC↑ | SRM AC↑ | Type | Name | LIF AC↑ | SRM AC↑ | Type | Name | LIF AC↑ | SRM AC↑ |
|---|---|---|---|---|---|---|---|---|---|---|---|---|---|---|---|
| Normal | SCNN | 99.42% | 99.13% | FGSM | SCNN | 66.33% | 56.85% | PGD | SCNN | 84.31% | 67.53% | 8-bit Loihi weight | SCNN | **98.86%** | 98.05% |
| | DT1 [24] | 99.40% | 99.05% | | DT1 [24] | 43.70% | 43.19% | | DT1 [24] | 77.32% | 61.82% | | DT1 [24] | 98.75% | 98.69% |
| | DT2 [26] | 98.24% | 98.13% | | DT2 [26] | 36.48% | 37.90% | | DT2 [26] | 78.25% | 60.44% | | DT2 [26] | 97.17% | 96.68% |
| | BDETT | **99.45%** | **99.15%** | | BDETT | **69.14%** | **57.01%** | | BDETT | **85.74%** | **68.06%** | | BDETT | **98.86%** | **98.22%** |
| GN (0, 0.3) | SCNN | 81.98% | 78.24% | GN (0, 0.5) | SCNN | 39.84% | 45.32% | 20% Zero weight | SCNN | 90.52% | 95.10% | 30% Zero weight | SCNN | 84.37% | 89.75% |
| | DT1 [24] | 56.19% | 54.38% | | DT1 [24] | 39.92% | 41.13% | | DT1 [24] | 87.20% | 93.58% | | DT1 [24] | 80.25% | 83.09% |
| | DT2 [26] | 33.80% | 26.23% | | DT2 [26] | 14.35% | 9.81% | | DT2 [26] | 79.43% | 83.00% | | DT2 [26] | 66.77% | 70.19% |
| | BDETT | **85.09%** | **78.68%** | | BDETT | **47.74%** | **46.34%** | | BDETT | **96.37%** | **96.59%** | | BDETT | **90.68%** | **91.02%** |

for LIF and SRM models. Furthermore, we use the original fitted parameters in our DTT, and Eq. 5 becomes $T_i^l(t + 1) = 1.0 + 10e^{\frac{-(v_i^l(t+1) - v_i^l(t))}{3}}$. For obstacle avoidance, with the originally fitted constants, the LIF-based policy cannot produce any successful pass even under the standard testing condition; SR drops from 92.5% to 0%. For the HalfCheetah-v3 and Ant-v3 tasks, with the originally fitted constants, the rewards achieved by a LIF-based policy dropped from 11064 to $-35$ and 5276 to $-9$, respectively. Note that an untrained BDETT-based policy achieves $-124$ and $-73$ rewards for these two continuous control tasks. Image classification tasks follow the same pattern; that is, with the originally fitted constants, the Top-1 accuracy achieved by a LIF-based policy dropped from 99.45% to 9.80%. These experimental results validate that the proposed statistical cues are essential to the proposed method.

## 5 Conclusion

This work introduces a novel biologically inspired BDETT scheme to SNNs that significantly improves generalization, and as such, fills a gap between biological research and machine learning. Dynamic threshold behavior plays an essential role in maintaining a neuronal homeostasis in biological nervous systems. Motivated by this observation, we propose a dynamic threshold scheme to achieve homeostasis in artificial SNNs. We assess the proposed approach in real-world tasks under normal and severely degraded conditions to validate its generalization capabilities. We find that the proposed dynamic threshold achieves strong homeostasis along with generalization to diverse degraded conditions. This finding is a step toward employing bioplausible SNNs in real-world applications. As future work, we plan to implement the proposed scheme on neuromorphic hardware to broadly deploy BDETT in future robotic platforms.

## Acknowledgements

This work was supported in part by National Key Research and Development Program of China (2022ZD0210500, 2018AAA0102003, 2021ZD0112400), the National Natural Science Foundation of China under Grant 61972067/U21A20491/U1908214, and the Innovation Technology Funding of Dalian (2020JJ26GX036). Felix Heide was supported by an NSF CAREER Award (2047359), a Sony Young Faculty Award, a Project X Innovation Award, and an Amazon Science Research Award.

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
