# OpenReview forum: "Biologically Inspired Dynamic Thresholds for Spiking Neural Networks"
_NeurIPS.cc/2022/Conference — NeurIPS 2022 Accept_

### Official Review · Reviewer_r3Sy · 2022-07-04

**Rating:** 5
**Confidence:** 3
**Soundness:** 2 fair
**Presentation:** 2 fair
**Contribution:** 2 fair

**Summary:**

The authors propose a dynamic spiking threshold function that consists of DET and DTT. DET is determined by the distributions of the thresholds and membrane potentials over the neurons in a given layer at a given timestep. DET is reconfigured s.t. the larger the average potential over the in-layer neurons, the larger the threshold, so that it avoids too high firing rate over the neurons as well as it induces competition among the neurons. DTT effectively realizes a refractory period s.t. it raises the spiking threshold when the potential decreases. The effect of the proposed spiking threshold (BDETT) applied to SNN-based policy functions for reinforcement learning on two tasks, i.e., robot obstacle avoidance and robotic continuous control. The results highlight a better performance for the use of BDETT in terms of success rate and homeostasis than the baseline methods.

**Questions:**

Q1. Applicability of BDETT to other domains. It is required to apply the proposed method to different application domains, e.g., computer vision. The goal is two-fold: (i) the impact of BDETT on SNNs can be highlighted, and (ii) true baseline performances are available, particularly, for vision domain, there exist tons of previous publications reporting their official performances. Can the authors evaluate the impact of BDETT on other application domains?

Q2. Complexity analysis. Can the authors evaluate the computation and space complexities of the proposed method and compare with previous works?


**Limitations:**

The authors did not address the limitations of the present work. But, I believe that the current result limits the impact of the present work to merely two simple tasks with unreliable baseline methods.

**Strengths And Weaknesses:**

$\textbf{Strengths}$:

S1. BDETT highlights a better performance than the baseline methods at least for a given set of parameters.
S2. As the authors highlight, BDETT has bio-plausible grounds and is novel.

$\textbf{Weaknesses}$:

W1. Weakness of baseline. the baseline performance is questionable given that the performance of BDETT is not compared with the performance of any previously published works. I am convinced that BDETT outperforms the previous methods that the authors addressed for a given set of parameters. However, I am not convinced if the proposed baseline is the ground-truth or close to the ground-truth.

W2. Wrong citations. I found that Refs. 24 and 26 are irrelevant to the baseline methods DT1 and DT2.

W3. Lack of in-depth analysis of BDETT. I am slightly tired of the authors’ emphasis on the bio-plausibility of BDETT. I agree on the point that bio-plausibility of a newly proposed method is good. But, such bio-plausibility does not justify the proposed method. The authors took a biological notion and recreated the notion largely, so that I do not think the fidelity of BDETT to biological notions is very high. Instead, I would like to suggest the authors to systematically address the advantages of BDETT over the previous methods from an engineering viewpoint rather that bio-plausibility.

W4. Additional computation and space complexity. The evaluation and memorization of BDETT for all neurons cause additional computation and space complexity, which should have been addressed in detail.

---

> ### Author Response · Authors · 2022-08-02
> **Response to Reviewer r3Sy Part (1/3)**
>
> Thank you for your insightful feedback and suggestions!
>
> >**Q1: Weakness of baseline. the baseline performance is questionable given that the performance of BDETT is not compared with the performance of any previously published works. I am convinced that BDETT outperforms the previous methods that the authors addressed for a given set of parameters. However, I am not convinced if the proposed baseline is the ground-truth or close to the ground-truth.**
>
> We compared BDETT to four state-of-the-art methods published in the last two years. Specifically, we compare static threshold methods, SAN[9] and PopSAN[35], and two recent dynamic schemes, DT1[24] and DT2[26].
>
> Specifically, SAN and PopSAN are two reinforcement learning methods that do not require preexisting ground truth mappings. These methods learn behavior by experiencing rewards for actions, which is distinctively different from supervised learning from a training set of labeled examples provided by a knowledgable external supervisor. [R4-1]
>
> >**Q2: Wrong citations. I found that Refs. 24 and 26 are irrelevant to the baseline methods DT1 and DT2.**
>
> DT1 is defined by Eqs. 4 and 5 of Hao et al.[24], on page 8. The $\alpha$ in Eq. 5 of Hao et al.[24] was set to 1.0 as the maximum value of the increment is 1.0 in our experimental settings. DT2 is defined by Eq. 4 of Kim et al.[26], on page 5.
>
> >**Q3: Bio-plausibility of BDETT. I agree on the point that bio-plausibility of a newly proposed method is good. But, such bio-plausibility does not justify the proposed method. The authors took a biological notion and recreated the notion largely, so that I do not think the fidelity of BDETT to biological notions is very high. Instead, I would like to suggest the authors to systematically address the advantages of BDETT over the previous methods from an engineering viewpoint rather that bio-plausibility.**
>
> BDETT is a *bio-inspired* algorithm (also reflected in the title) with the *aim* of a bio-plausible behavior, i.e., homeostasis. BDETT aims to achieve homeostasis, keeping the firing rates constant regardless of the external conditions. To this end, we deliberately deviate from bio-plausible dynamic threshold schemes adapt to the SNN models; see also manuscript lines 149-151, "the proposed dynamic energy threshold is inspired by this biological predictive model but includes several changes that are critical for the model to be effective in SNNs." We will further highlight this in the abstract.
>
> We provide an analysis of the relationship between the two main components of the method, DET and DTT, in section _Interaction of DET and DTT_ and Q1 of reviewer wR8D. We provide a systematic evaluation for all tested tasks from an engineering viewpoint in Section 4. Specifically, for each tested task we reported the following conventional metrics established in the respective engineering sub-domain:
>
> |      Task          | Performance metric    |
> |------------        |-----------------------|
> | Obstacle avoidance | Success Rate (SR): percentage of successful passes out of 200 trials |
> |                    |overtime percentage (OTP): overtime is defined as a trial in which the robot cannot reach the goal within 1000 steps but does not touch any obstacle.
> | HalfCheetha-V3     | Reward                 |
> | Ant-v3             | Reward                 |

---

> > ### Author Response · Authors · 2022-08-02
> > **Response to Reviewer r3Sy Part (2/3)**
> >
> > >**Q4: Applicability of BDETT to other domains. It is required to apply the proposed method to different application domains, e.g., computer vision. The goal is two-fold: (i) the impact of BDETT on SNNs can be highlighted, and (ii) true baseline performances are available, particularly, for vision domain, there exist tons of previous publications reporting their official performances. Can the authors evaluate the impact of BDETT on other application domains?**
> >
> > In principle, the proposed BDETT may be used for any SNN-based method. We apply the method to three robotic tasks under 19 different experimental setups, including normal and degraded conditions. Note also that the proposed BDETT has been tested with two different SNN models, i.e., LIF and SRM.
> >
> > We agree that applying the proposed method to different application domains is interesting. As the reviewer suggested, we applied the proposed BDETT to image classification as an established computer vision task.
> >
> > To this end, we adopted the SCNN model [R3-1] and train on the MNIST dataset. Following the experimental setup of [R3-1], each pixel of an MNIST image is encoded into 30 Poisson spikes as inputs to SCNN for both training and testing.
> >
> > Similar to our experimental setup for robotic control tasks, we also designed two different degraded conditions: degraded inputs and weight uncertainty.
> >
> > * **Adversarial samples as degraded inputs**: to test the generalization on degrated inputs we use existing adversarial attack methods to generate relevant degraded inputs for the image classification tasks.
> >     * ‘FGSM $\epsilon=x$’: Fast gradient sign method (FGSM) [R3-2] with $\epsilon=x$;
> >     * ‘PGD $iter_\epsilon=x$ $iter_{num}=y$’: projected gradient descent (PGD) [R3-3], with iteration epsilon of x and iteration number of y for each attack step.
> > * **Weight uncertainty**: We also evaluate the robustness to internal weight uncertainty as follows.
> >     * ‘GN(0, x)’: adding Gaussian noise (GN) with zero mean and standard deviation of x to all synaptic weights;
> >     * ‘x% zero weight’: for the synaptic weights between every two adjacent layers, we randomly set x% of them to 0;
> >
> >
> > *Directly applying the proposed approach without any changes to image classification in degraded conditions compares favorably across all experimental settings and in terms of generalization.* With stronger degradations, the top 1 classification accuracy of both baseline and our approach decreases, but the proposed method is less affected, validating BDETT for this vision task.
> >
> > | Experimental setup                                 | Threshold Type  | LIF-based SCNN    | SRM-based SCNN    |
> > |--------------------------------------|-------|--------|--------|
> > | Original                                | Static Threshold  | 99.42% | 99.13% |
> > |                                      | BDETT | **99.45%** | **99.15%** |
> > | FGSM $\epsilon$=0.20                                | Static Threshold  | 66.33% | 56.85% |
> > |                                      | BDETT | **69.14%** | **57.01%** |
> > | PGD $iter_\epsilon$=0.01 $iter_{num}$=20                                 | Static Threshold  | 84.31% | 67.53% |
> > |                                      | BDETT | **85.74%** | **68.06%**
> > | GN(0, 0.3)                           | Static Threshold  | 81.98% | 78.24% |
> > |                                      | BDETT | **85.09%** | **78.68%** |
> > | GN(0, 0.5)                           | Static Threshold  | 39.84% | 45.32% |
> > |                                      | BDETT | **47.74%** | **46.34%** |
> > | 20% zero weight                      | Static Threshold  | 90.52% | 95.10% |
> > |                                      | BDETT | **96.37%** | **96.59%** |
> > | 30% zero weight                      | Static Threshold  | 84.37% | 89.75% |
> > |                                      | BDETT | **90.68%** | **91.02%** |

---

> > > ### Author Response · Authors · 2022-08-02
> > > **Response to Reviewer r3Sy Part (3/3)**
> > >
> > > >**Q5: Complexity analysis. Can the authors evaluate the computation and space complexities of the proposed method and compare with previous works?**
> > >
> > > **Runtime complexity:**
> > >
> > > The computational complexity of the proposed BDETT is bounded by the computational complexity of calculating the mean, maximum, and minimum, i.e., Eqs. 3, 4, and 6. Therefore, the upper bound of estimating BDETT complexity, $\Theta_i^l(t+1)$, is $O(n)$, where $n$ is the number of neurons on the $l$-th layer.
> > >
> > > Other methods, DT1 and DT2, are bounded by the summation operations, and their upper bound are also $O(n)$, where $n$ is also the number of neurons on a layer; see Eqs. 8 and 9 in Supplementary Note 2.
> > >
> > > We report the layer-wise running time with PyTorch 1.2 on an i7-7700 CPU and NVIDIA GTX 1080Ti GPU. As we can see the running time of the proposed BDETT for the testing network is 1.36 ms.
> > >
> > > |            | Layer 1 (256 neurons) | Layer 2 (256 neurons) | Layer 3 (256 neurons) | Layer 4(2 neurons) | Total |
> > > |------------|-----------------------|-----------------------|-----------------------|--------------------|-------|
> > > | DET (ms)   | 0.18                  | 0.19                  | 0.19                  | 0.18               | 0.74  |
> > > | DTT (ms)   | 0.11                  | 0.11                  | 0.11                  | 0.10               | 0.43  |
> > > | BDETT (ms) | 0.34                  | 0.35                  | 0.35                  | 0.32               | 1.36  |
> > >
> > > **Memory complexity:**
> > >
> > > To evaluate BDETT, $\Theta_i^l(t+1)$, we need to evaluate $V_m^l(t)$, $V_{\theta}^l(t)$, and $\mu(\Theta_i^l(t))$. Therefore, the upper bound of the memory complexity is $O(n)$, where $n$ is the number of neurons on the $l$-th layer. The lower bound is $O(1)$.
> > >
> > > DT1 and DT2 offer the same memory complexity.
> > >
> > > >**Limitations: The authors did not address the limitations of the present work. But, I believe that the current result limits the impact of the present work to merely two simple tasks**
> > >
> > > We discussed the limitations of our work in lines 80-83 of the main manuscript. The three tasks we use to assess the proposed BDETT have not been tackled successfully with SNNs under different degraded conditions. In contrast, the majority of SNN works still leverage classification tasks on small datasets for their evaluation and only under normal conditions.
> > >
> > >
> > > #### References
> > > [R4-1] Sutton, R. S., & Barto, A. G. (2018). Reinforcement learning: An introduction. MIT press.

---

> > > > ### Author Response · Authors · 2022-08-06
> > > > **Response to Reviewer r3Sy**
> > > >
> > > > Dear reviewer r3Sy,
> > > >
> > > > Addressing your questions and concerns, we conducted additional experiments on image classification as an established computer vision task. The results are well aligned with our findings in the three robotic tasks; the proposed threshold method improves the generalization of this vision task. Besides, we provided more details and insights into our baselines; and we provide the requested analysis of the time and space complexity.
> > > >
> > > > In light of this, we would like to know whether the experimental results and updated exposition have addressed your concerns. If so, we hope you would be willing to increase your score. We appreciate the effort that went into reviewing our work!
> > > >
> > > > The Authors

---

> > > > > ### Comment · Reviewer_r3Sy · 2022-08-09
> > > > > **Response**
> > > > >
> > > > > I appreciate the authors' effort on the response and additional experiments. I raise my rate accordingly.

---

### Official Review · Reviewer_T4kg · 2022-07-04

**Rating:** 7
**Confidence:** 3
**Soundness:** 3 good
**Presentation:** 3 good
**Contribution:** 3 good

**Summary:**

This paper introduces a new bioinspired dynamic energy-temporal threshold (BDETT) scheme for spiking neural networks (SNNs), and
validates the strong homeostasis along with its generalization to diverse degraded conditions. Experiments are conducted on the robot obstacle avoidance and continuous control tasks. The result shows that BDETT outperforms existing static and heuristic threshold approaches.  The supplemental video is impressive.


**Questions:**

See the weakness part.

**Limitations:**

None.

**Strengths And Weaknesses:**

Strenghts:
1. Introduces a bioinspired dynamic threshold scheme for SNNs that increases their generalizability.
2, Designs a new method that uses layerwise statistical cues of SNNs to set the parameters of our bioinspired threshold method.
3. The proposed threshold scheme achieves bioplausible homeostasis, dramatically enhancing the generalizability across tasks, including obstacle avoidance and robotic control, and in normal and degraded conditions.


Weakness:
1. The paper combines two dynamic thresholds that exhibit positive and negative correlations with the average membrane potential. It would be better to explain the reason and the related mathematical formulation in detail.  Plus, the author compares BDETT with four variants of the spiking actor-network (SAN), are there any other recent SNNs to compare with?
2. The experiments are restricted to the robot and control tasks. It would be better to include more tasks such as tasks in computer vision. 3. The supplementary material has a heavy overlap with the main body part.

---

> ### Author Response · Authors · 2022-08-02
> **Response to Reviewer T4kg Part (1/2)**
>
> Thank you for the insightful feedback and suggestions!
>
> >**Q1: The paper combines two dynamic thresholds that exhibit positive and negative correlations with the average membrane potential. It would be better to explain the reason and the related mathematical formulation in detail.**
>
> The positive and negative correlations in the proposed method are motivated by Fontaine et al.[16] who found that the spike threshold was positively correlated with the average membrane potential preceding spikes and negatively correlated with the rate of depolarization. We emphasize that DET leverages the _magnitude of the membrane potential_ to estimate a threshold, while the DTT is based on the _preceding rate of depolarization_. Eqs. 2-4 provide mathematical formulations for DET, also illustrated in Figure 1b. Eqs. 5-6 formalize DTT along with illustrations in Figure 1c.
>
> >**Q2: The author compares BDETT with four variants of the spiking actor-network (SAN), are there any other recent SNNs to compare with?**
>
> For the obstacle avoidance tasks, we compared SRM- and LIF-based SAN and SAN-NR, four variants of SAN[9]. For the continuous robot control tasks, we compared SRM- and LIF-based PopSAN, two variants of PopSAN[35]. Note that both SAN and PopSAN are pure SNNs, meaning they have no ANN/CNN-based components. To the best of our knowledge, SAN and PopSAN are the only relevant pure SNN-based models in the reinforcement learning domain.

---

> > ### Author Response · Authors · 2022-08-02
> > **Response to Reviewer T4kg Part (2/2)**
> >
> > >**Q3: The experiments are restricted to the robot and control tasks. It would be better to include more tasks such as tasks in computer vision.**
> >
> > In principle, the proposed BDETT may be applied to tasks solved with an  SNN. We assess the method on *three tasks under 19 different experimental setups, including normal and degraded conditions.* The proposed BDETT has been tested with two different SNN models, i.e., LIF and SRM.
> >
> > Nevertheless, we conducted additional experiments on image classification. To this end, we adopted the SCNN model [R3-1] and trained on the MNIST dataset. Following the experimental setup of [R3-1], each pixel of an MNIST image is encoded into 30 Poisson spikes as inputs to SCNN for both training and testing.
> >
> > Similar to our experimental setup for robotic control tasks, we also designed two different degraded conditions: degraded inputs and weight uncertainty.
> >
> > * **Adversarial samples as degraded inputs**: to test the generalization on degrated inputs we use existing adversarial attack methods to generate relevant degraded inputs for the image classification tasks.
> >     * ‘FGSM $\epsilon=x$’: Fast gradient sign method (FGSM) [R3-2] with $\epsilon=x$;
> >     * ‘PGD $iter_\epsilon=x$ $iter_{num}=y$’: projected gradient descent (PGD) [R3-3], with iteration epsilon of x and iteration number of y for each attack step.
> > * **Weight unvertainty**: We also evaluate the robustness to internal weight uncertainty as follows.
> >     * ‘GN(0, x)’: adding Gaussian noise (GN) with zero mean and standard deviation of x to all synaptic weights;
> >     * ‘x% zero weight’: for the synaptic weights between every two adjacent layers, we randomly set x% of them to 0;
> >
> >
> > *Directly applying the proposed approach without any changes to image classification in degraded conditions compares favorably across all experimental settings and in terms of generalization.* With stronger degradations, the top 1 classification accuracy of both baseline and our approach decreases, but the proposed method is less affected, validating BDETT for this vision task.
> >
> > | Experimental setup                                 | Threshold Type  | LIF-based SCNN    | SRM-based SCNN    |
> > |--------------------------------------|-------|--------|--------|
> > | Original                                | Static Threshold  | 99.42% | 99.13% |
> > |                                      | BDETT | **99.45%** | **99.15%** |
> > | FGSM $\epsilon$=0.20                                | Static Threshold  | 66.33% | 56.85% |
> > |                                      | BDETT | **69.14%** | **57.01%** |
> > | PGD $iter_\epsilon$=0.01 $iter_{num}$=20                                 | Static Threshold  | 84.31% | 67.53% |
> > |                                      | BDETT | **85.74%** | **68.06%**
> > | GN(0, 0.3)                           | Static Threshold  | 81.98% | 78.24% |
> > |                                      | BDETT | **85.09%** | **78.68%** |
> > | GN(0, 0.5)                           | Static Threshold  | 39.84% | 45.32% |
> > |                                      | BDETT | **47.74%** | **46.34%** |
> > | 20% zero weight                      | Static Threshold  | 90.52% | 95.10% |
> > |                                      | BDETT | **96.37%** | **96.59%** |
> > | 30% zero weight                      | Static Threshold  | 84.37% | 89.75% |
> > |                                      | BDETT | **90.68%** | **91.02%** |
> >
> >
> > #### References
> > [R3-1] Wu, Y., Deng, L., Li, G., Zhu, J., & Shi, L. (2018). Spatio-temporal backpropagation for training high-performance spiking neural networks. Frontiers in neuroscience, 12, 331.
> >
> > [R3-2] I. J. Goodfellow, J. Shlens, and C. Szegedy, “Explaining and harnessing adversarial examples,” arXiv preprint arXiv:1412.6572, 2014.
> >
> > [R3-3] Madry, A. et al. “Towards Deep Learning Models Resistant to Adversarial Attacks.” ArXiv abs/1706.06083 (2018): n. pag.

---

### Official Review · Reviewer_wR8D · 2022-07-05

**Rating:** 8
**Confidence:** 3
**Soundness:** 3 good
**Presentation:** 3 good
**Contribution:** 3 good

**Summary:**

In this work, the authors aim at bridging this gap by introducing a novel bioinspired dynamic energy-temporal threshold (BDETT) scheme for spiking neural networks (SNNs). Meanwhile, the authors propose a BDETT scheme mirrors two bioplausible observations: a dynamic threshold has 1) a positive correlation with the average membrane potential and 2) a negative correlation with the preceding rate of depolarization.
--------------------------------------------------------------------------------------------------
The author's revisions address all my concerns and I would further recommend this manuscript.

**Questions:**

1. Since the authors claim in the methods section that the two dynamic threshold mechanisms can help each other achieve optimal settings. Thus, more sufficient evidence needs to be provided to support this viewpoint.

**Limitations:**

The authors emphasize in the introduction section that it still requires hardware engineering efforts.

**Strengths And Weaknesses:**

Strenghts:
1. The authors designed a bio-inspired dynamic threshold mechanism to enhance the generalization of SNNs.
2. The new dynamic energy-temporal threshold mechanism reflects the two biological observations.
3. The parameter setting of the dynamic threshold mechanism can use layerwise control of statistical information.

Weaknesses:
1. The interaction of DET and DTT does not seem to be supported by ablation experiments.

---

> ### Author Response · Authors · 2022-08-02
> **Response to Reviewer wR8D**
>
> Thank you for the insightful feedback and suggestions!
>
> >**Q1: Since the authors claim in the methods section that the two dynamic threshold mechanisms can help each other achieve optimal settings. Thus, more sufficient evidence needs to be provided to support this viewpoint.**
>
> In the section _Interaction of DET and DTT_ in our manuscript, we illustrate the interaction between DET and DTT with two examples. As suggested by the reviewer, we provide additional experimental evidence in the following.
>
> #### Interaction DET/DTT with low potential fluctuations.
>
> * **Experimental setup**: We randomly chose a timestamp and recorded all postsynaptic membrane potentials and spiking thresholds. Then, for each layer, we randomly selected $X$ neurons based on the binomial distribution with a probability of 0.5. The chosen neurons were added random positive noise, generated based on a normal distribution $\mathcal{N}(0.2, 0.05)$. The mean of 0.2 is around 20% of the average of the recorded membrane potentials. To reduce the impact of the randomness, we did 5-round tests and reported the average and standard deviation (STD) of the obtained DETs and DTTs.
> * **Experimental thesis**:  In this case, we expect DET increases as the noise increases the membrane potential. DTT should remain at a relatively constant threshold (i.e., a + 1) as the preceding rate of depolarization caused by the noise is close to 0.
> * **Experimental result**: The layerwise mean $(\mu)$ and STD $(\sigma)$ of the 5-round DETs and DTTs with and without added noise are reported below, aligning well with the experimental thesis.
>
> |            | $\mu(X)$ | original ($\mu$ / $\sigma$)    | with added noise ($\mu$ / $\sigma$)    |
> |------------|-----|--------------|---------------------------|
> | layer 1 DET|130.4|1.4950 / 0.0051    |1.5143 / 0.0066            |
> | layer 1 DTT|     |0.0570 / 0.0064    |0.0571 / 0.0063            |
> | layer 2 DET|128.4|2.1548 / 0.0120    |2.1745 / 0.0111            |
> | layer 2 DTT|     |0.2725 / 0.0154    |0.2724 / 0.0152            |
> | layer 3 DET|126.6|3.3528 / 0.0788    |3.3720 / 0.0777            |
> | layer 3 DTT|     |0.4549 / 0.0033    |0.4549 / 0.0033            |
>
> #### Interaction DET/DTT with fast membrane potential drop
>
> * **Experimental setup**: We adopted the same binomial distribution as in the first experiment and randomly selected $X$ neurons. To mimic fast membrane potential drops from $t$ to $t+1$, we added random negative membrane potentials with a larger magnitude than the first experiment, which was generated by sampling a normal distribution $\mathcal{N}(-2.0, 0.5)$.
> * **Experimental thesis**: In this scenario, even though DET decreases with the reduced membrane potential, we expect DTT to increase faster, and BDETT to increase the overall threshold.
> * **Experimental result**: The layerwise mean and STD of the 5-round $X$ DETs, DTTs, and BDETTs with and without fast membrane potential drop are shown in the table below. Again, the findings align with the experimental thesis.
>
>
> |                | $\mu(X)$ |original ($\mu$ / $\sigma$)    | fast potential drop ($\mu$ / $\sigma$) |
> |----------------|-----|-------------------|---------------------------|
> | layer 1 DET    |128.2|1.4919 / 0.0089    |1.4032 / 0.0106            |
> | layer 1 DTT    |     |0.0579 / 0.0044    |1.0115 / 0.0251            |
> | layer 1 BDETT  |     |0.7749 / 0.0039    |1.2074 / 0.0124            |
> | layer 2 DET    |124.8|2.1865 / 0.0292    |2.0348 / 0.0514            |
> | layer 2 DTT    |     |0.2863 / 0.0053    |1.1938 / 0.0332            |
> | layer 2 BDETT  |     |1.2364 / 0.0148    |1.6143 / 0.0293            |
> | layer 3 DET    |130.4|3.6456 / 0.1114    |3.3802 / 0.1298            |
> | layer 3 DTT    |     |0.4479 / 0.0141    |1.3337 / 0.0307            |
> | layer 3 BDETT  |     |2.0468 / 0.0610    |2.3570 / 0.0574            |

---

### Official Review · Reviewer_qTH8 · 2022-07-13

**Rating:** 5
**Confidence:** 4
**Soundness:** 2 fair
**Presentation:** 3 good
**Contribution:** 2 fair

**Summary:**

The authors study spiking neural networks (SNNs) with a biologically-motivated dynamic spiking threshold.  The underlying hypothesis is that having a dynamic threshold allows the network to fire similarly with a wide array of stimuli / external conditions, allowing the network to potentially generalize across such conditions better.  They show that the networks increase robots' performance under degraded conditions, and increase homeostasis per metrics they provide, such as the mean and standard deviation of the firing rates across trials.

**Questions:**

1- What is the motivation for choosing the 3 statistics outlined for homeostatsis? While the mean and standard deviation of the firing rates are 2 natural statistics to measure, why they (and the 3rd) should be the ones to choose for measuring homeostasis is unclear. Are there drawbacks in deciding on these particular ones (e.g., perhaps in choosing these, the networks perform worse wrt other candidate metrics?) ?

2 - Do the shaded regions in Figs 2e and 3d represent SDs, SEMs, or something different?

2- L234: "soft-reset" is unclear

**Limitations:**

Yes

**Strengths And Weaknesses:**

**Originality:**

To my knowledge, dynamic thresholds have not been studied in the context of ANNs.

**Quality:**

The authors have provided biologically motivated models and intuitions for why these models might be beneficial in ANNs.  They have then provided highly applied use cases to test the performance of their models.  However, while the performance of the networks is illustrated, no statistical tests are undertaken.  The reader is thus left to wonder as to whether which, if any, of the performance improvements are statistically significant.  Moreover, while the authors have shown both an increase in homeostasis with their metrics and an increase in performance in 2 robotic tests, the reader is left with only this correspondence.  Presumably there are ways in which generalization can occur without improved homeostasis, and ways to increase homeostasis too far, resulting in decreased learning performance?  Yet these possibilities are not raised in the present work.

**Clarity:**

Overall, the motivation, intuitions, experimental setups, and analyses are all clear. However, several aspects remain unclear to me.  Please see questions, below.  Also, it is only made clear in the appendix that the robotic experimental setup is indeed a virtual one, from what I have been able to see.  The reader's understanding would be facilitated if this information were included in the main text and in Fig 2's caption.


**Significance:**

Biological neurons are known, as pointed out by the authors, to have dynamic thresholds, though the function of such thresholds is, as yet, uncertain. This study represents a welcome introduction of these known dynamics into ANNs, where such functionality can be queried in the context of feedforward networks, and the authors provide an existence proof that in certain contexts, such dynamics might contribute to increased generalization performance.  However, without stronger statistical analysis and, especially, a substantively deeper exploration of how homeostasis and generalization are linked beyond example cases, it is unclear how robust the authors' findings are, and whether the hypothesized link from increased homeostasis --> increased generalization performance is causal, and under what circumstances if so.

---

> ### Author Response · Authors · 2022-08-02
> **Response to Reviewer qTH8 part (1/2)**
>
> Thank you for the insightful feedback and suggestions!
>
> **Scope and correlation between homeostasis and generalization.**
>
> We report empirical evidence that the bio-plausible state of homeostasis can allow for strong generalization across diverse tasks. However, we do not provide guarantees for this generalization capability across task domains but rather make a first step in demonstrating that bio-plausible homeostasis *can* improve generalization. Although such theoretical results are out of the scope of our work, we hope that this work provides an impulse in this direction.
>
> Specifically, we establish a direct connection between homeostasis and generalization for three different robotic tasks, i.e., obstacle avoidance, HalfCheetah-v3, and Ant-v3, with two widely used SNN models, i.e., LIF and SRM, for two different types of degradations:
>
> &emsp;1. Measurement Uncertainty (i.e., degraded inputs)
>
> &emsp;2. Internal Changes (i.e., weight uncertainty)
>
> For each type of degradation, three different experimental settings were conducted. Our experimental results validate that our approach offers homeostasis and generalization in *all 19 different experimental settings with two different SNN models (i.e., LIF and SRM) and two different timestamp settings (i.e., T=5 and T=25); total 19x2x2=76 experiments. For each SNN model and timestamp setting, we conducted seven experiments for the obstacle avoidance task; six and six for HalfCheetah-v3 and Ant-v3, respectively.* All experimental settings are listed in the table below.
>
>
> |                   |                                               | Obstacle Avoidance | HalfCheetah-v3 | Ant-v3 |
> |-------------------|-----------------------------------------------|--------------------|----------------|--------|
> | Dynamic obstacle  |                                               | $\surd$            |                |        |
> | Degraded inputs   |0.2 (set the range of the 3rd, 9th, and 15th lasers to 0.2 m)                                            | $\surd$            |                |        |
> |                   |6.0 (set the range of the 3rd, 9th, and 15th lasers to 6.0 m)                                             | $\surd$            |                |        |
> |                   |GN<br>$(clip(s_{input} + \mathcal{N}(0, 1.0), 0.2, 6.0))$| $\surd$            |                |        |
> |                   |Random joint position                          |                    | $\surd$        |$\surd$ |
> |                   |Random joint velocity                          |                    | $\surd$        |$\surd$ |
> |                   |GN<br>$(s_{input} + \mathcal{N}(0, 1.0))$                |                    | $\surd$        |$\surd$ |
> | Weight uncertainty|8-bit Loihi weights                            | $\surd$            | $\surd$        |$\surd$ |
> |                   |GN weights                                     | $\surd$            | $\surd$        |$\surd$ |
> |                   |30% zero weights                               | $\surd$            | $\surd$        |$\surd$ |
>
>
> >**The reader's understanding would be facilitated if this information (simulation-based testing) were included in the main text and in Fig 2's caption.**
>
> We agree with the reviewer, and we will add this information along with Figure 2.
>
>
>
> >**Q1: What is the motivation for choosing the 3 statistics outlined for homeostasis? While the mean and standard deviation of the firing rates are 2 natural statistics to measure, why they (and the 3rd) should be the ones to choose for measuring homeostasis is unclear. Are there drawbacks in deciding on these particular ones (e.g., perhaps in choosing these, the networks perform worse wrt other candidate metrics?) ?**
>
> The three statistics are motivated by existing work investigating homeostasis in biological neural networks. Specifically, Zenke et al.[45] pointed out that "homeostasis comprises any compensatory mechanism that stabilizes neural firing rates in the face of plasticity induced changes.” Lazar et al.[44] define it as an “...effective way of modeling the effect of a network of inhibitory interneurons that maintains a constant level of firing in the network.” Turrigiano et al.[R1-1] find “the ability of neurons to adjust synaptic or intrinsic excitability in a homeostatic manner to keep firing rates relatively constant.” As such, together, the three statistics reflect the constantness of the firing rates of an SNN-based network.

---

> > ### Author Response · Authors · 2022-08-02
> > **Response to Reviewer qTH8 part (2/2)**
> >
> > >**Q2: Do the shaded regions in Figs 2e and 3d represent SDs, SEMs, or something different?**
> >
> > The shaded regions in Figs 2e and 3d represent SDs. We will clarify this in the revised version.
> >
> > >**Q3: L234: "soft-reset" is unclear**
> >
> > "Soft-reset" means that when the membrane potential passes a threshold, it is reset by subtracting the threshold value. We adopt this concept from Tang et al. [R1-2]. In contrast, "Hard-reset" implies that once the membrane potential is over a threshold, the potential is reset to zero. We will clarify this in our revised version.
> >
> > #### References
> >
> > [R1-1] Turrigiano, G. G., & Nelson, S. B. (2004). Homeostatic plasticity in the developing nervous system. Nature reviews neuroscience, 5(2), 97-107.
> >
> > [R1-2] Tang, G., Kumar, N., Yoo, R., & Michmizos, K. P. (2020). Deep reinforcement learning with population-coded spiking neural network for continuous control. arXiv preprint arXiv:2010.09635.

---

> > > ### Author Response · Authors · 2022-08-06
> > > **Response to Reviewer qTH8**
> > >
> > > Dear reviewer qTH8,
> > >
> > > Addressing your review, we summarized all 76 conducted experiments, which all empirically confirm that homeostasis improves SNN generalization. As such, we believe our work has the potential to ignite excitement in this research direction. In addition, we discuss the choices of the three statistics we reported for homeostasis from a bio-plausible perspective.
> > >
> > > In light of this, we would like to know whether the provided analysis and insights have addressed your concerns. If so, we hope you would be willing to increase your score. Thank you for your consideration!
> > >
> > > The Authors

---

> > > > ### Comment · Reviewer_qTH8 · 2022-08-09
> > > > **Increasing score**
> > > >
> > > > Hello,
> > > >
> > > > I thank the authors for their responses.  The results of their experiments outlined in the table are compelling.  I would still suggest the authors have better statistical results, but I will increase my score to a 5.

---

### Author Response · Authors · 2022-08-02
**This is a top-level reply, please see our answer to each individual reviewer thread for details**

We thank all reviewers for their thoughtful feedback and we are happy to see the positive reception. All reviewers agree that the proposed BDETT is a novel threshold scheme. In particular, reviewer qTH8 finds “dynamic thresholds have not been studied in the context of SNNs”, and reviewer r3Sy mentioned “BDETT has bio-plausible grounds and is novel.” The reviewers agreed that “the proposed threshold scheme achieves bioplausible homeostasis”[T4kg] and can “enhance the generalization of SNNs”[wR8D]. We address remaining concerns as separate responses to the the reviewers.

---

### Meta-Review · Area_Chair_aKrV · 2022-08-24

**Recommendation:** Accept
**Confidence:** Less certain

**Metareview:**

The paper proposes a biologically plausible dynamic thresholding mechanism. Spiking neural nets with dynamic thresholding appears to be novel. The paper does a good job of motivating the choice of the model and illustrating its benefits across a series of control tasks. All reviewers support the acceptance of the paper conditional on the following points to be included in the revised manuscript:

- The new experiments on image processing performed during the discussion phase have to be included in the revised version.
- I agree with Reviewer r3Sy that the paper overly emphasizes the biological plausibility of the method as a point of strength. Please try to focus the paper more on the technical benefits and analysis of the proposed method, following the instructions of the reviewer.
- Include the complexity analysis of the model in the revised version.
- Please import some of the tables provided during the rebuttal to the revised manuscript.
- Explicitly denote the details of the statistics of your experimental results. It might be helpful to import some of the tables from the supplementary materials to the main text.

I recommend the acceptance of this paper.

**Award:**

No

---

### Decision · Program_Chairs · 2022-09-14

Accept